# Evidences for Mutant Huntingtin Inducing Musculoskeletal and Brain Growth Impairments via Disturbing Testosterone Biosynthesis in Male Huntington Disease Animals

**DOI:** 10.3390/cells11233779

**Published:** 2022-11-25

**Authors:** Libo Yu-Taeger, Arianna Novati, Jonasz Jeremiasz Weber, Elisabeth Singer-Mikosch, Ann-Sophie Pabst, Fubo Cheng, Carsten Saft, Jennifer Koenig, Gisa Ellrichmann, Taneli Heikkinen, Mahmoud A. Pouladi, Olaf Riess, Huu Phuc Nguyen

**Affiliations:** 1Department of Human Genetics, Faculty of Medicine, Ruhr University Bochum, 44801 Bochum, Germany; 2Institute of Medical Genetics and Applied Genomics, University of Tuebingen, 72076 Tuebingen, Germany; 3Department of Neurology, Ruhr University Bochum, 44801 Bochum, Germany; 4Charles River Discovery Services, 70210 Kuopio, Finland; 5Department of Physiology, Yong Loo Lin School of Medicine, National University of Singapore, Singapore 117597, Singapore; 6British Columbia Children’s Hospital Research Institute, Department of Medical Genetics, University of British Columbia, Vancouver, BC V5Z 4H4, Canada; 7Center for Rear Disease [ZSE], University of Tuebingen, 72076 Tuebingen, Germany

**Keywords:** Huntington disease, musculoskeletal growth, brain growth, sex-difference, BACHD rats, R6/2 mice, testosterone synthesis, transcription dysregulation

## Abstract

Body weight (BW) loss and reduced body mass index (BMI) are the most common peripheral alterations in Huntington disease (HD) and have been found in HD mutation carriers and HD animal models before the manifestation of neurological symptoms. This suggests that, at least in the early disease stage, these changes could be due to abnormal tissue growth rather than tissue atrophy. Moreover, BW and BMI are reported to be more affected in males than females in HD animal models and patients. Here, we confirmed sex-dependent growth alterations in the BACHD rat model for HD and investigated the associated contributing factors. Our results showed growth abnormalities along with decreased plasma testosterone and insulin-like growth factor 1 (IGF-1) levels only in males. Moreover, we demonstrated correlations between growth parameters, IGF-1, and testosterone. Our analyses further revealed an aberrant transcription of testosterone biosynthesis-related genes in the testes of BACHD rats with undisturbed luteinizing hormone (LH)/cAMP/PKA signaling, which plays a key role in regulating the transcription process of some of these genes. In line with the findings in BACHD rats, analyses in the R6/2 mouse model of HD showed similar results. Our findings support the view that mutant huntingtin may induce abnormal growth in males via the dysregulation of gene transcription in the testis, which in turn can affect testosterone biosynthesis.

## 1. Introduction

Huntington disease (HD) is a fatal neurodegenerative disorder caused by an expanded polyglutamine tract in the N-terminal region of the huntingtin (HTT) protein [1], which is expressed ubiquitously throughout the body with the highest levels in the brain and testis [2]. Clinically, HD is mainly characterized by motor deficits, cognitive impairments, and psychiatric disturbances due to central nervous system dysfunctions [3]. However, since HTT is expressed in all cells, a wide range of peripheral alterations and symptoms have also been described in HD, including weight loss, cardiomyopathy, sexual dysfunction, as well as skeletal muscle pathology [4]. Patients with HD suffer from muscle atrophy and dysfunction and present muscular nuclear inclusion forms [4]. In line with changes in HD patients, HD mouse models display muscle atrophy and muscle functional deficits, as well as metabolic alterations, nuclear inclusions, and transcriptional dysregulation in skeletal muscle [5,6,7,8].

Weight loss is frequently found in HD patients and HD animal models [9]. Several studies in HD patients showed that among different body composition parameters, lean body mass was mostly reduced [10], while results on body fat were inconsistent, suggesting that musculoskeletal mass is an important factor contributing to the body weight (BW) deficits in HD patients. Body weight in HD rodents correlates with the expression levels of full-length HTT. In particular, HD rodent models carrying more copies of full-length HTT, such as BACHD mice and YAC mice, display increased body weight [11,12,13,14], whereas models expressing a fragment of mutant HTT (mHTT) or HD knock-in models show decreased body weight due to the depletion of full-length wild type HTT (wtHTT) [15,16,17,18,19]. Lower body weight and body mass index (BMI) are already present in patients at an early disease stage [20,21] as well as in HD mutation carriers starting in childhood [22]. Similarly, reduced bone and muscle mass were observed at one month of age in BACHD transgenic rats, an HD rat model expressing full-length human mHTT [23,24]. All this evidence may indicate that BW loss and reduced BMI in HD are an abnormal growth event rather than an effect of cell degeneration and may result from altered developmental processes. In support of this, huntingtin has been suggested to play essential roles during development, such as control of mitosis and neurogenesis [25,26], as well as regulation of synaptogenesis and spine structure [27]. Moreover, HD has been shown to be associated with neurodevelopmental alterations [28,29,30].

Interestingly, regardless of increased or decreased body weight, many HD rodent models show sex differences in the body weight phenotype, and this has been observed across rat and mouse models carrying different transgenic constructs. BACHD females show more prominent body weight gain compared to BACHD males, while a more pronounced body weight loss has been observed in HD males compared to HD females in tg51 rats and R6/2, HdhQ111, zQ175, and CAGKI mice [17,18,31,32,33,34,35], as well as in a humanized mouse model of HD [19]. Furthermore, a growing number of studies revealed that BW and BMI are more severely affected in males than in females, in both HD patients [10] and in HD animal models [18,36], indicating a sex-specific effect of the HD mutation.

In the growth process, the growth hormone (GH) is a major regulator whose release is controlled by growth hormone-releasing hormone (GhRH) and somatostatin, which are in turn secreted by the hypothalamus [37]. In the liver, GH promotes the synthesis of insulin-like growth factor 1 (IGF-1), another important hormone participating in the regulation of cell growth and proliferation. Furthermore, sex steroids, especially testosterone, interact with GH and enhance IGF-1′s response to GH [38]. Testosterone can also interact with IGF-1 controlling the production of IGF-binding proteins (IGFBP) [39], which can modulate IGF-1 bioavailability by regulating its circulation half-life. Notably, circulating testosterone levels were shown to be reduced in male HD patients and male R6/2 mice that also present BW loss and muscle atrophy [40,41]. All these results suggest a potential association between testosterone and BW reduction in HD, especially during growth.

While the above-mentioned evidence for BW alterations, growth impairments, hormonal changes and sex differences in HD have been separately reported in different studies on patients and animal models, it remains unclear to what extent all these factors are interconnected. To better understand the link between growth alterations and sex steroids, we evaluated longitudinal BW changes as well as musculoskeletal and brain growth in male and female BACHD (TG5) transgenic rats expressing full-length mutant human huntingtin, and wild-type (WT) littermates at 1.5, 3, and 7 months of age (representative of young, young adult, and middle-aged adult age groups, respectively). In parallel, we determined the circulating levels of GH, IGF-1, and testosterone, as well as the correlation between each of these hormones and growth parameters. We further analyzed the mRNA expression of genes related to testosterone biosynthesis in order to understand the potential causes of testosterone reduction in HD. To corroborate our findings, we investigated main findings in the R6/2 mouse model expressing human HTT exon 1 [42].

## 2. Materials and Methods

### 2.1. Animals

All rats and mice were housed and treated in accordance with the German and the Finnish Animal Welfare Acts, and the guidelines of the Federation of European Laboratory Animal Science Associations based on the European Union legislation (Directive 2010/63/EU). All experiments were approved by the local ethics committees (Regierungspraesidium Tuebingen, Germany; Regierungspraesidium Bochum, Germany and State Provincial Office of Southern Finland). All animals were housed under standard conditions (12:12-h light-dark cycle; water and food ad libitum). The animals in each experiment were littermates. When assigning the animals from the litters to the experimental groups, we balanced the animals from each litter among the experimental groups. During each measurement, animals of different groups were randomized according to age, sex, and genotype. All experiments were performed by experimenters blind to the experimental groups.

### 2.2. Tissue Harvest and Gross Dimensional Measurements

After decapitation, the entire brain was removed promptly from the skull, while the gastrocnemius was carefully separated from the tibia and fibula bones at the level of the hind limbs. Brain and gastrocnemius from each rat were immediately weighed after collection (accuracy = 0.0001 g). Tibia length was measured using a digital caliper ruler (accuracy = 0.01 mm). Trunk blood was collected in EDTA tubes (BD Vacutainer, Franklin Lakes, NJ, USA; No. 368861) upon sacrifice.

### 2.3. Estrous Cycle Monitoring

Estrous cyclicity was assessed via vaginal smear cytology. One week before sacrifice, vaginal smears were collected once a day for 5 days to follow the cycle phases over time. Sterile cotton swabs were wetted in distilled water and gently inserted in the vagina of female rats in order to collect cells from the vaginal lumen and walls. Vaginal samples were then placed on a histology glass and observed under a standard laboratory light microscope (40×). The estrus phase of the cycle was recognized by the predominance of anucleate cornified cells in unstained cell material [43].

### 2.4. Plasma Preparation

Blood samples were centrifuged at 5000× *g* for 5 min at 4 °C within 30 min after sacrifice to obtain plasma. Plasma aliquots were stored at −80 °C until use.

### 2.5. Enzyme-Linked Immunosorbent Assay (ELISA)

The plasma concentrations of IGF-1, testosterone, 17β-estradiol, GH and luteinizing hormone (LH) were quantified using Mouse/Rat IGF-1 Quantikine ELISA kit (MG100, R&D Systems, Wiesbaden, Germany), Testosterone ELISA kit (ADI-900-065, Enzo Life Sciences, Farmingdale, NY, USA), 17β-estradiol high sensitivity ELISA kit (ADI-900-174, Enzo Life Sciences), Rat/Mouse Growth Hormone ELISA kit (EZRMGH-45k, Millipore, Darmstadt, Germany) and Rat Luteinizing Hormone ELISA kit (MBS764675, MyBioSource, San Diego, CA, USA), respectively. Analyses with each kit were performed according to the manufacturer’s instruction.

### 2.6. Ex Vivo T2-MRI

Before imaging, all brains were rinsed with saline and embedded in perfluoropolyether (FOMBLIN). T2-weighted MRI was performed using a horizontal 7 T magnet with an inner bore diameter of 160 mm (Magnex Scientific Ltd., Oxford, UK) and equipped with an actively shielded Magnex gradient set (max. gradient strength: 400 mT/m, bore 100 mm) interfaced to a Varian DirectDrive console. Linear RF volume-coil was used for transmission and surface-phased array coil for receiving (Rapid Biomedical GmbH, Rimpar, Germany).

For determination of total brain, striatal and cortical volumes, T2-weighted continuous multi-slice images covering the whole brain (number of slices, 21) were acquired using fast spin-echo sequence with TR = 4500 ms, echo train length ETL = 4, effective TE = 36 ms, a matrix size of 512 × 256, FOV of 30 × 30 mm^2^, and a slice thickness of 0.6 mm. The average of 4 measurements was taken. The coronal images were analyzed for total brain, striatal, and cortical volumes using an in-house written analysis program run under MATLAB (The MathWorks Inc, Natick, MA, USA) environment.

### 2.7. Protein Extraction and Western Blotting

After dissection, tissues were kept at –80 °C till use. To prepare protein lysates, tissues were homogenized with an Ultra-Turrax T10 (VWR) in 10 volumes (*w*/*v*) of ice-cold modified RIPA buffer (50 mM Tris pH 8.0, 150 mM sodium chloride, 1.0% NP-40, 0.5% sodium deoxycholate, 0.1% SDS, 5 mM EDTA) supplied with Complete Proteinase Inhibitor Cocktail tablets (1873580, Sigma-Aldrich) and PhosSTOP phosphatase Inhibitor Cocktail tablets (0490683701, Sigma-Aldrich). After a further 5-min sonication step in an ultrasonic bath for shearing genomic DNA, the lysates were centrifuged at 16,200× *g* and 4 °C for 20 min to obtain the soluble protein.

Western blot analyses were performed as described earlier [44]. Primary antibodies were diluted in TBS-T (TBS with 0.1% Tween 20) and the respective dilution factors are summarized in Table A1. For total protein detection, membranes were incubated with SYPRO Ruby Protein Blot Stain (Thermo Fisher Scientific) according to the manufacturer’s protocol, prior to blocking. For caspase-3 analyses, a commercial sample of cell extracts treated with cytochrome c served as positive control (9663, Cell signaling). All fluorescence signals were detected and quantified using the ODYSSEY FC Imaging System with Image Studio software version 4.0 (both LI-COR Biosciences, Bad Homburg, Germany).

### 2.8. RNA Isolation and Reverse Transcription-Quantitative PCR

Total RNA was extracted from testis and ovary samples using RNeasy Plus Universal Midi Kit (73442, Qiagen, Hilden, Germany). Reverse transcription was performed with QuantiTect Reverse Transcription Kit (218061, Qiagen). Quantitative real-time PCR was performed using QuantiTect SYBR Green PCR Kit (204145, Qiagen) on a LightCycler 480 instrument and software (Roche). Expression was calculated relative to the mean of the WT group after normalization to the geometric mean relative expression of three reference genes (Actb, Rsp18 and Rsp29 for the testis of male rats, GAPDH, Rsp29, and Srp14 for the ovary of female rats, and GAPDH, Rsp18, and Hprt1 for the testis of mice), which were equally expressed between transgenic and WT animals and displayed smallest variance within 8 examined reference genes. Primer sequences are listed in Table A2.

### 2.9. Histochemical and Immunohistochemical Staining

For the assessment of brain morphology and cAMP activation in testis, rats were transcardially perfused with 4% paraformaldehyde in phosphate-buffered saline (pH 7.4), followed by overnight post-fixation of the brains in the same fixative. For immunohistochemistry, rat brains were embedded in one gelatin block, and 40-μm coronal sections were freeze-cut and collected into 24 series (NeuroScience Associates, Knoxville, TN, USA). One of these series (containing 1 every 24 sections) was used for morphological analysis with anti-NeuN antibody (1:300, MAB377, Merck). Free-floating staining was performed as previously described [23]. Testes were collected and fixed as previously described [45]. Paraffin sections were stained with anti-cAMP (ab134902, abcam) at a dilution of 1:1000 and secondary goat anti-rabbit antibody (1:1000, BA-1000, Vector Laboratories). For muscle staining, rat gastrocnemii were rapidly frozen in liquid nitrogen directly after dissection, then stored at −80 °C till sectioning. Cross-sections with 8-µm thickness were stained using a Trichrome Stain Kit (ab150686, Abcam).

### 2.10. Statistical Analysis

All statistical analyses were performed using GraphPad Prism 9.1.2. The results of gross dimensional measurements and plasma levels of hormones in each age group were analyzed by regular two-way ANOVA with Sidak’s post-hoc test, while repeated measures two-way-ANOVA followed by Turkey’s post-hoc test was used for the longitudinal analysis of body weight. Correlation coefficients between two variables were tested by simple linear regression analysis. Two-tailed unpaired Student’s *t*-test were used for the comparison of body composition in females at 12 months of age, protein levels in the muscles and testes, and mRNA levels in the testes and ovaries, as well as for the comparison of HTT expression between testis and ovary. Analyses of HTT expression in peripheral organs of male rats were performed using one-way ANOVA analysis with Dunnett’s post-hoc test. *p* values smaller than or equal to 0.05 were regarded as statistically significant.

## 3. Results

### 3.1. Reduced Musculoskeletal Growth in Male, but Not in Female BACHD Rats

Our previous study on body composition in male BACHD rats up to 12 months showed decreased muscle and bone weight (from three months onwards), but increased adipose tissue weight (from six months onwards) [24]. To know if the body composition in female BACHD rats changes in a similar manner, we measured abdominal white adipose tissue (WAT) weight and gastrocnemius weight in 12-month old female rats. In line with previous results in male BACHD rats, WAT weight was significantly increased (F_(1,34)_ = 9.592, *p* = 0.0029). However, gastrocnemius weight did not differ between genotypes in female rats (Figure 1A). To confirm the difference in muscle weight loss between female and male BACHD rats, and to assess at what time point it starts, we performed a longitudinal measurement of body weight and a transversal analysis of gastrocnemius weight and tibia length in female and male WT and BACHD rats. The BW was analyzed at six time points from six to 30 weeks of age. Two-way repeated measures ANOVA detected an increased BW in female BACHD rats (genotype effect: F_(1,30)_ = 7.954, *p* = 0.0084) and a decreased BW in male BACHD rats (genotype effect: F_(1,30)_ = 6.733, *p* = 0.0145) compared with WT littermates of the same sex (Figure 1B). Gastrocnemius weight and tibia length were measured as growth parameters at three different ages: young (1.5 months), young adult (3 months), and middle-aged adult (7 months) (Figure 1C,D). The statistical analyses of growth parameters were performed separately for each age group. Two-way ANOVA detected a main effect of genotype for gastrocnemius weight (F_(1,59)_ = 39.39, *p* < 0.0001) and for tibia length (F_(1,59)_ = 15.16, *p* = 0.0003) in the middle-aged adult group. We also found a genotype–sex interaction for gastrocnemius weight in the young adult group (F_(1,59)_ = 7.398; *p* = 0.0091) and in the middle-aged adult group (F_(1,59)_ = 25.23, *p* < 0.0001), as well as a genotype–sex interaction for tibia length in the middle aged-adult group (F_(1,59)_ = 13.58, *p* = 0.0005). Sidak’s post-hoc test demonstrated highly significantly reduced gastrocnemius weight (Figure 1C) and tibia length (Figure 1D) in males (all *p* < 0.0001), but not in females, in the middle-aged group. A smaller and less significant gastrocnemius weight reduction in males was found in the young adult group as well (*p* = 0.0125) (Figure 1C). Overall, these results show a sex-specific musculoskeletal growth reduction in male BACHD rats, which was strongest in the middle-aged group.

### 3.2. Increased Activation of Proteins Regulating Cell Growth

We questioned whether the reduced musculoskeletal mass in BACHD rats is triggered by organ atrophy or a consequence of impaired growth. To this end, we evaluated muscle fiber degeneration. To assess muscle morphology, the gastrocnemii of BACHD rats from the middle-aged adult group were stained with trichrome stain which allows the visualization of muscle fibers, collagen tissue and nuclei. We did not observe evidence for muscle degeneration such as muscle fibers loss, abnormal collagen volume fraction, or altered central nuclei in female or male BACHD rats at 7 months of age. These results indicate that the sex differences in muscle and brain mass reduction in BACHD rats are unlikely to be dependent on atrophy.

The mechanistic target of rapamycin (mTOR) is the master regulator of cell growth [46]. S6 ribosomal protein (S6RP) kinase (S6K) and 4E-BP1 are the best-known substrates of mTOR, and phosphorylation (activation) of S6RP and 4E-BP1 promote protein translation and cell proliferation [47,48]. To evaluate the growth state in BACHD rats, we therefore assessed the phospho-S6RP and phospho-4E-BP1 as well as their total protein expression levels in the gastrocnemius muscle of rats at 7 months of age. Student *t*-test analyses showed that both S6RP and 4E-BP1 have comparable expression levels in BACHD rats and WT littermates, but their phosphorylated forms differed between genotypes with an opposite direction of change in females and males (Figure 2B). Interestingly, sex differences in the phosphorylation of S6RP and 4E-BP1 matched those observed in the musculoskeletal mass, showing decreased phospho-S6RP (*p* = 0.0071) and phospho-4E-BP1 (*p* = 0.0369) in male BACHD rats but increased phospho-S6RP (*p* = 0.0438) and unchanged phospho-4E-BP1 (*p* = 0.2486) in female BACHD rats. The ratio between phosphorylated protein and total protein showed similar results. Altogether, our results indicate that the reduction in musculoskeletal mass in male BACHD rats is associated with altered growth mechanisms.

### 3.3. Decreased Plasma Levels of IGF-1 and Testosterone in Male, but Not in Female BACHD Rats

IGF-1 is a potential factor that may explain the sex-differences in growth observed in BACHD rats. We compared the plasma levels of IGF-1 between genotypes and sexes within each age group. Two-way ANOVA demonstrated a significant genotype x sex interaction in the middle-aged group (F_(1,60)_ = 9.602, *p* = 0.0030). Sidak’s post-hoc revealed a significantly lower plasma level of IGF-1 in male BACHD rats compared to male WT littermates (*p* = 0.0073), but no genotype difference in plasma IGF-1 levels in females (Figure 3A). Plasma IGF-1 levels correlated significantly with gastrocnemius weight in both sexes (female: F_(1,30)_ = 4.480, *p* = 0.0427; male: F_(1,30)_ = 8.929, *p* = 0.0056 (Figure 3B) and with tibia length in male rats (*p* = 0.0003) (Figure 3C). A significant difference between the slopes of the regression lines in females and males was detected in the regression analysis between plasma IGF-1 levels and tibia length (F_(1,60)_ = 0.4595, *p* = 0.05). These results may indicate that abnormal plasma IGF-1 levels influence the growth of BACHD rats, particularly in males.

Since testosterone, 17β-estradiol and GH play essential roles in regulating musculoskeletal growth either directly or indirectly via IGF-1 modulation [37,39], we measured plasma levels of these hormones in the middle-aged group, which displayed male-specific reductions in musculoskeletal growth parameters and IGF-1. A significant genotype effect as well as a highly significant interaction between genotype and sex were detected only for testosterone levels (genotype effect: F_(1,59)_ = 14.64, *p* = 0.0003; genotype × sex interaction: F_(1,59)_ = 15.49, *p* = 0.0002). Sidak’s post-hoc test revealed significantly lower testosterone levels in male BACHD rats compared to male WT littermates (*p* < 0.0001), but not in female BACHD rats in comparison with female WT littermates (Figure 3D). There was no difference between genotypes in plasma 17β-estradiol or GH within each sex (Table 1). To assess if there was a correlation in each of the three investigated hormones with IGF-1 or with musculoskeletal growth parameters, further regression analyses were applied. Results showed that in male rats, testosterone levels correlated positively with IGF-1, gastrocnemius weight and tibia length (IGF-1: F_(1,30)_ = 11.34, *p* = 0.0022; gastrocnemius: F_(1,30)_ = 17.79, *p* = 0.0002; tibia: F_(1,30)_ = 5.251, *p* = 0.0300) (Figure 3E–G), while 17β-estradiol levels were negatively correlated with each of the two growth parameters. In females, there was no correlation in any of the sex steroids either with IGF-1 or with growth parameters. The plasma levels of GH showed no correlation in males, and only a positive correlation with the length of tibia in females (F_(1,30)_ = 4.676, *p* = 0.0390) (Table 1). The results of these analyses indicate a possible association between reduced testosterone plasma levels and growth impairment in male BACHD rats.

### 3.4. Reduction of Brain Mass in Female and Male BACHD Rats

As our results revealed sex-dependent differences in musculoskeletal growth in BACHD rats, we examined whether there was a similar effect on the brain, which is most affected in HD. Brain weights were compared between sexes and genotypes within each age group (Figure 4A). Two-way ANOVA analysis demonstrated highly significant main effects of sex (young group: F_(1,60)_ = 22.98, young adult group: F_(1,56)_ = 41.91, middle-aged group: F_(1,59)_ = 38.08; all *p* < 0.0001) and genotype (young group: F_(1,60)_ = 22.34, young adult group: F_(1,56)_ = 32.07, middle-aged group: F_(1,59)_ = 95.65; all *p* < 0.0001) at all three investigated ages. Unlike for the peripheral organs, brain weight was reduced in both male (young group, *p* = 0.0004; both young and middle-aged adult groups, *p* < 0.0001) and female (young group, *p* = 0.0186; young adult group, *p* = 0.0033; middle-aged adult group, *p* < 0.0001) BACHD rats compared to WT littermates of the same sex. However, a genotype–sex interaction was still found in the middle-aged group similar to the musculoskeletal parameters (F_(1,59)_ = 5.128, *p* = 0.0272). To validate these results, we estimated the whole brain volume using ex vivo magnetic resonance imaging (MRI) in a cohort of 14 months old BACHD rats and WT littermates (Figure 4B). Two-way ANOVA revealed a highly significant whole brain volume reduction in BACHD rats compared to WT littermates (genotype effect: F_(1,30)_ = 35.64, *p* < 0.0001), and a significantly smaller whole brain volume in female rats compared to male rats (sex effect: F_(1,30)_ = 23.67, *p* < 0.0001). Consistent with the brain weight results, two-way ANOVA analyses for whole brain volume also showed a significant interaction between sex and genotype (F_(1,30)_ = 6.401, *p* = 0.0169). These data show that the reductions in the brain weight and volume of BACHD rats differ between sexes. Regression analyses revealed that brain weight significantly correlated with IGF-1 and testosterone plasma levels in males (F_(1,30)_ = 14.05, *p* = 0.0182) (Figure 4C,D). To evaluate whether the sex differences in brain volume change in BACHD rats could be attributed to a different degree of neurodegeneration in females and males, neuronal degeneration and apoptosis were investigated. Coronal brain slices of BACHD and WT rats at nine months of age were stained with neuronal nuclei antibody. The results showed comparable brain structure, cell density and neuronal morphology in BACHD rats and same-sex WT littermates (Figure A1A). We also analyzed the activities of caspase 3 and caspase 6 which are involved in apoptosis processes and have been shown to be increased in mHTT transgenic animals [49,50]. Using western blot analysis, cleaved (activated) caspase 3 and caspase 6 were compared between BACHD rats and WT littermates. Until nine months of age, no activated form of caspase 3 or caspase 6 was detected in BACHD or WT rats. A representative image of caspase 3 is shown in Figure A1B.

### 3.5. No Difference in HTT Expression between Female and Male BACHD Rats

It has been reported that the expression level of full-length HTT modulates IGF-1 level in YAC and BACHD mice in a sex-mixed group [13]. We therefore questioned whether female and male BACHD rats have different expression levels of the transgenic and endogenous HTT protein. Using western blot analysis, mHTT and wtHTT protein expression levels were compared between female and male BACHD rats in brain and muscle (Figure 5A). Protein expression levels of mHTT and wtHTT did not differ between sexes in the middle-agedadult group.

### 3.6. Huntingtin Expression Is Most Abundant in Brain and Testis in BACHD Rats

In humans and mice, HTT was shown to be expressed throughout the body with highest expression levels in brain and testis [2,14]. To gain knowledge on the HTT expression across organs in rats, we quantified mHTT and wtHTT in different organs of male BACHD transgenic rats, using western blot analysis. One-way ANOVA analysis followed by Dunnett’s post-hoc test showed that mHTT expression was significantly higher in the brain compared to any of the peripheral organs. The testes displayed the most abundant expression of wtHTT with significantly higher levels than in all other organs whereas the brain showed significantly higher wtHTT expression than muscle, lung, heart, liver, spleen, and kidney. The expression levels of mHTT and wtHTT in the testis were at least two-fold higher than in any of the other peripheral organs. In addition, the expression level of mHTT and wtHTT was compared between ovary and testis in BACHD transgenic rats. While mHTT levels did not differ significantly between sexual organs, wtHTT levels were higher in testis than in ovary (Figure 5B).

### 3.7. Reduced Expression Levels of Genes Related to Steroid Biosynthesis in the Testes of Male BACHD Rats

Numerous studies showed that mHTT causes transcriptional dysregulation in HD humans and animal models [51,52,53,54], raising the question of whether the high mHTT expression level in the BACHD rat testis may decrease testosterone synthesis via altering gene transcription. We therefore investigated mRNA expression levels of genes related to steroid biosynthesis in the testes of middle-aged rats. Four of the twelve investigated genes showed decreased transcription levels in BACHD rats, i.e., cholesterol side-chain cleavage enzyme (*Cyp11a1*), scavenger receptor class b type 1 (*Scarb1*), 3-hydroxy-3-methylglutaryl-CoA synthase 1 (*Hmgcs1*), and methylsterol monooxygenase 1 (*Sc4mol*) (Figure 6A). The remaining genes *Pgds*, *3-Hsd*, *17-HSD*, *Scl7a5*, *Gst*, *Rlf*, *Hmgcr*, and *Sc5d* were unchanged (Figure A2). A trend towards reduced protein levels of CYP11A1 in the testes of middle-aged BACHD rats was observed using western blot analysis (Figure 6B). Besides high mHTT expression levels, an alternative possible explanation for the down-regulated gene expression in the testis is a reduced LH release because of a dysfunctional hypothalamic–pituitary axis, that is known in HD [55]. In the Leydig cells of the testis, this would result in a decreased activation of cyclic adenosine monophosphate (cAMP)/protein kinase A (PKA) signaling pathway which regulates gene transcription. We therefore analyzed the LH plasma protein level using ELISA and the Lhr gene mRNA expression in the testis with real-time PCR. Moreover, we quantified cAMP-positive cells on immunohistochemically stained testis slices, as well as the level of cAMP-activated catalytic subunit C of PKA (pPKA C) using western blot. None of these investigated factors showed a significant difference nor a trend between genotypes (Figure 6B–D). Additionally, we assessed the ovarian gene expression level of *Cyp11a1*, *Scarb1*, *Hmgsc1*, and *Sc4mol*, which like in the testis, are regulated by the LH-promoted cAMP/PKA signaling pathway. In contrast to the testis, the gene expression level of *Cyp11a1* was higher in the ovary of BACHD rats compared to WT littermates. All the other genes assessed in the ovary, showed comparable expression levels between genotypes (Figure 6E). Taken together, our results demonstrate that key genes involved in steroid biosynthesis are downregulated in the testis, with no change in the LH-promoted cAMP/PKA pathway.

### 3.8. Reduced Expression Levels of Genes Related to Steroid Biosynthesis in the Testes of Male BACHD Rats

The sex differences in altered growth and testosterone biosynthesis observed in BACHD rats were further validated in the R6/2 mouse model which is the most widely used animal model of HD. R6/2 mice express the human exon 1 HTT with approximately 140 CAG repeats [42] and display a robust and progressive neurological phenotype replicating many clinical features of HD patients [56]. BW of R6/2 mice were measured from five to 11 weeks of age. At 11 weeks of age, we analyzed circulating levels of testosterone in plasma, mRNA expression levels of steroid biosynthesis-related genes in testicular mRNA extracts, as well as CYP11A1 protein expression levels and cAMP-dependent PKA activation levels in testicular protein extracts. Longitudinal BW analyses using two-way repeated measures ANOVA indicated a main effect of genotype in male R6/2 mice vs. male WT mice (*p* = 0.0276), but no significant difference between female R6/2 and WT mice. A statistically significant BW reduction in male R6/2 mice started from 9 weeks of age onwards, when the BW gain in female R6/2 mice stopped (Figure 7A). A dramatic reduction in the plasma levels of testosterone in male R6/2 mice was revealed by an ELISA assay (*p* = 0.0011) (Figure 7B). Consistent with the results in BACHD rats, the steroid biosynthesis-related genes *Cyp11a1* (*p* < 0.0001), *Scarb1* (*p* = 0.0434), and *Hmgcs1* (*p* = 0.0068) were downregulated in R6/2 mice. *Cyp11a1* expression displayed a 94% reduction (Figure 7C) and its downregulation was confirmed at the protein level (Figure 7D) using western blot analyses, which showed a decrease to 12% of the level in WT littermates (*p* < 0.0001). Different than in BACHD rats, the levels of cAMP-activated subunit C of PKA in testis were significantly lower in R6/2 mice than in WT littermates (*p* = 0.0420) (Figure 7D).

## 4. Discussion

In this study, we showed sex differences in growth in both BACHD rats and R6/2 mice, which represent full-length and fragment animal models of HD, respectively. BW loss and muscle atrophy have been frequently reported in HD despite adequate nutrition. The underlying mechanisms can be partially explained by an altered energy balance [55]. However, the effect of sex differences in body weight change observed in HD patients and animals [6,12,13] as well as the sex-specific effects of the HTT gene on the brain volume development in children [57,58] indicate that there may be additional pathogenic mechanisms in the disease. In male BACHD rats, but not in female littermates, we found decreased circulating levels of testosterone hormones that positively correlated with growth parameters and IGF-1 circulating levels. Interestingly, the decreased testosterone levels were also shown in male HD patients [18]. Here, we provided evidence that mHTT induces growth impairment specifically in male HD animals by disturbing the transcription of testosterone biosynthesis-related genes.

It has been shown that testosterone deficiency is associated with decreased IGF-1 [39,59], while testosterone administration increases serum levels of IGF-1 [60], suggesting that decreased circulating IGF-1 levels in male BACHD rats may result from lower testosterone levels. Both IGF-1 and testosterone play important roles in regulating the growth of the skeletal muscle and bone [61,62]. Moreover, an increasing number of studies demonstrated that IGF-1 exerts neurotrophic and neuroregenerative effects at the brain level by regulating energy homeostasis, as well as neural and stem cell proliferation, differentiation, and survival [63,64]. Consistently, our male BACHD rats that displayed decreased circulating IGF-1 levels also showed impaired musculoskeletal growth and a larger reduction in brain weight and brain volume than female BACHD littermates.

Beyond central nervous system growth, both IGF-1 and testosterone could affect neuronal processes relevant to other aspects of HD pathogenesis. IGF-1 is able to restore mitochondrial function [65], regulate neuroendocrine function [66], modulate neuronal firing [67], and restore age-related impairments in dopaminergic neuronal function [68]. Testosterone can affect motor, cognitive, and motivational behaviors as well as dopamine synthesis in the substantial nigra [69,70,71]. In line with these behavioral effects, testosterone levels which were shown to be decreased in HD patients, negatively correlated with disease severity and dementia [41], suggesting an important role of this hormone in HD pathogenesis and symptomatology.

We questioned whether the sex differences in the BACHD rats’ musculoskeletal and brain growth could be attributed to a different expression level of wtHTT or mHTT between females and males in the affected organs. We were able to exclude this possibility as in both the brain and the gastrocnemius, wtHTT and mHTT expression levels did not differ between sexes. However, the testis in BACHD rats displayed the highest expression levels of mHTT and wtHTT among all peripheral organs, along with a significantly reduced expression of steroid biosynthesis-related genes, such as Cyp11a1, Scarb1, Hmgcs1, and Sc4mol. The same genes were also downregulated in R6/2 mice, suggesting that the observed gene expression changes in the testis are reproduced across animal models of HD.

Steroid hormones are synthesized from cholesterol and the initial enzymatic reaction in this process is catalyzed by Cyp11a. Scarb1 encodes for protein SR-B1, involved in the selective cholesterol uptake pathway. Both Hmgcs1 and Sc4mol encode for proteins essential in cholesterol synthesis. In this study, Cyp11a1 also displayed the most significant expression change among all examined genes and was downregulated at the protein level, pointing at a potentially important role of this gene in HD animals. In Leydig cells, the transcription of the gene Cyp11a1 is promoted by LH via the cAMP/PKA pathway. LH is released under stimulation of the HPG axis [72], and by binding to LHR induces the activation of cAMP-depended PKA which in turn regulates gene expression. Symptomatic 11-weeks old R6/2 mice in this study displayed a decreased level of the cAMP-activated subunit C of PKA in the testis, which could be the result of changes in LH levels and therefore related to HPG dysfunction. In support of this, a significant loss of GnRH producing neurons and HPG axis dysfunction, which could in turn affect the LH/cAMP/PKA pathway, were previously shown in symptomatic R6/2 mice between 9 and 12 weeks of age [40]. This evidence comprehensively suggests that HPG axis dysregulation may cause or contribute to the decreased testosterone biosynthesis in R6/2 mice at advanced disease stages. However, the question remains whether additional mechanisms are involved in this process. Different than in R6/2 mice, in BACHD rats, we observed decreased testosterone and altered transcription of testosterone biosynthesis-related genes despite no change in plasma LH levels or in the cAMP/PKA pathway, suggesting that HPG axis dysregulation is not necessary to cause testosterone impairment. As mHTT is known to disturb gene transcription, the high mHTT levels in the testis along with gene dysregulation hint at a possible direct effect of mHTT on gene transcription. Noteworthy, dysregulated expression of some testosterone biosynthesis-related genes has been shown in the brain of HD animals in several gene expression profiling studies. A downregulation of Cyp11a1 and Scarb1 in the cortex and Sc4mol in both striatum and cortex were reported in HD knock-in mice [73], decreased expression levels of Hmgcs1 and Sc4mol were found in the striatum of R6/2 mice with 150 CAG repeats [74], and a trend toward downregulation of Hmgcs1 was observed in the striatum of BACHD rats [75]. In female BACHD rats, we did not detect differences in the level of testosterone or 17β-estradiol, a female steroid partially synthesized in the ovary under HPG axis stimulation. Moreover, in the BACHD rat ovary, Cyp11a1 was significantly upregulated, while no change was found in the transcripts of the other three genes, which were down-regulated in the testis, indicating a downregulation of steroid-associated genes specifically in testis.

In conclusion, a sex-specific impaired growth in male HD animals may result from a decreased testosterone synthesis that in turn could be dependent on the reduced transcription of testosterone biosynthesis-related genes in the testis where the mHTT expression levels are the highest after the brain. As testosterone can affect the growth and development of different organs including the brain, our findings point to an important role of testosterone deficits in the HD pathogenesis. The effect of other sex specific factors, e.g., energy intake and exercise, or local mHTT effects on growth parameters cannot be completely ruled out. Modulating mHTT levels or testosterone levels in the testis in future studies will be determinant to establish a causal relationship between mHTT and the observed brain and muscle changes in the BACHD rat model.

Our results highlight the importance of taking HD carriers into account for diagnosis and treatment, as we showed that mHTT affected both body and brain growth, suggesting an effect of mHTT at early stages. Furthermore, the data underscore the importance of sex as a biological variable in HD diagnosis and therapy, supporting the testing of sex-specific hormone-based therapies for HD. Additionally, early reduced production of testosterone and its biosynthesis-related protein provide possibilities to develop new diagnostic biomarkers and therapeutic targets. Importantly, sex differences were also shown for different aspects in Alzheimer disease (AD) [76] and Parkinson disease (PD) [77]. Decreased circulating testosterone levels and testosterone deficiency have been reported in AD and PD patients, respectively [78,79], whereas research in rats showed the development of PD-like pathologies following castration [80]. Similar to HD patients, low testosterone levels correlated with cognitive deficits in patients with PD [81], while low testosterone levels in elderly men were associated with an increased risk for AD [82]. Altogether, this evidence indicates that sex steroids may play an important role in multiple neurodegenerative diseases.

## Figures and Tables

**Figure 1 cells-11-03779-f001:**
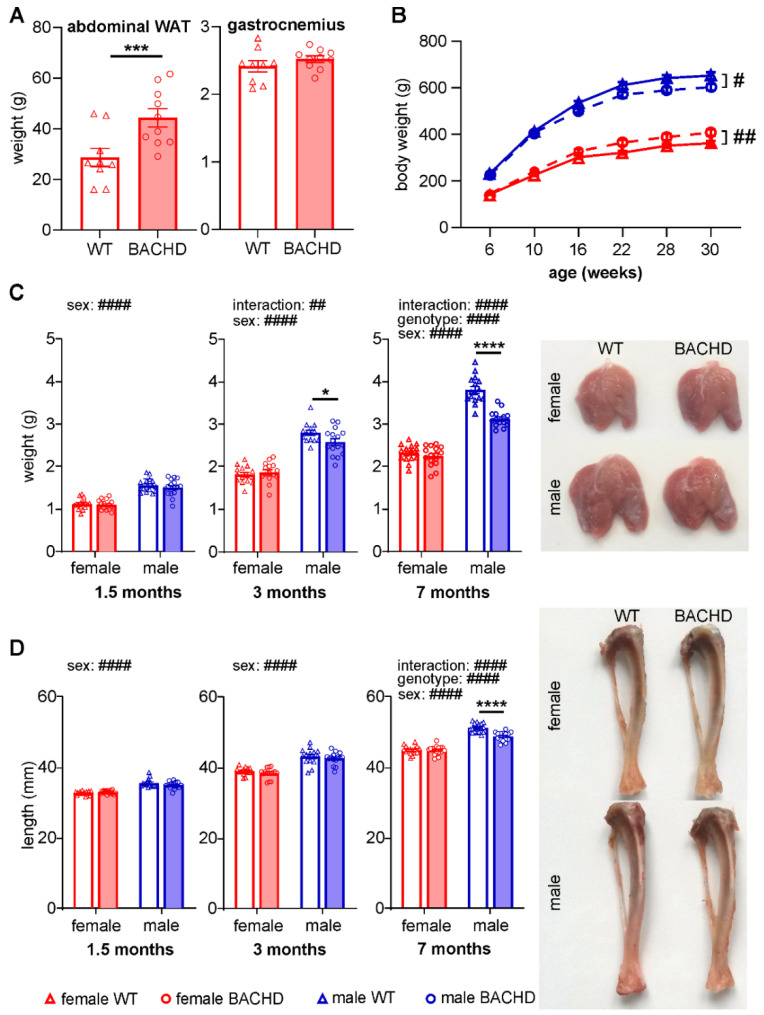
Sex-differences in body weight gain in BACHD rats and reduced musculoskeletal growth in male BACHD rats. (**A**) Abdominal white adipose tissue (WAT) and gastrocnemius weight in female BACHD rats and WT littermates at 12 months of age (*n* = 12/genotype). (**B**) Longitudinal analysis of body weight in female and male WT and BACHD rats from 6 to 30 weeks of age (*n* = 16/genotype/sex). (**C**) Gastrocnemius weight and (**D**) tibia length in three age groups: young (1.5 months), young adult (3 months) and middle-aged adult (7 months) (*n* = 12–16/genotype/sex/age). Right panels in C and D show representative images of gastrocnemius and tibia, respectively. Statistical significance was determined by two-tailed unpaired Student’s *t*-test in A, two-way repeated measures ANOVA followed by Tukey’s post-hos test in B, and regular two-way ANOVA followed by Sidak’s post-hoc test in C and D. Data are expressed as mean values ± SEM. # indicates the results of ANOVA analyses (**B**–**D**); * indicates the results of *t*-test (**A**) and post-hoc test (**C**,**D**). # *p* ≤ 0.05; ## *p* ≤ 0.01; *** *p* ≤ 0.001; ####/**** *p* ≤ 0.0001.

**Figure 2 cells-11-03779-f002:**
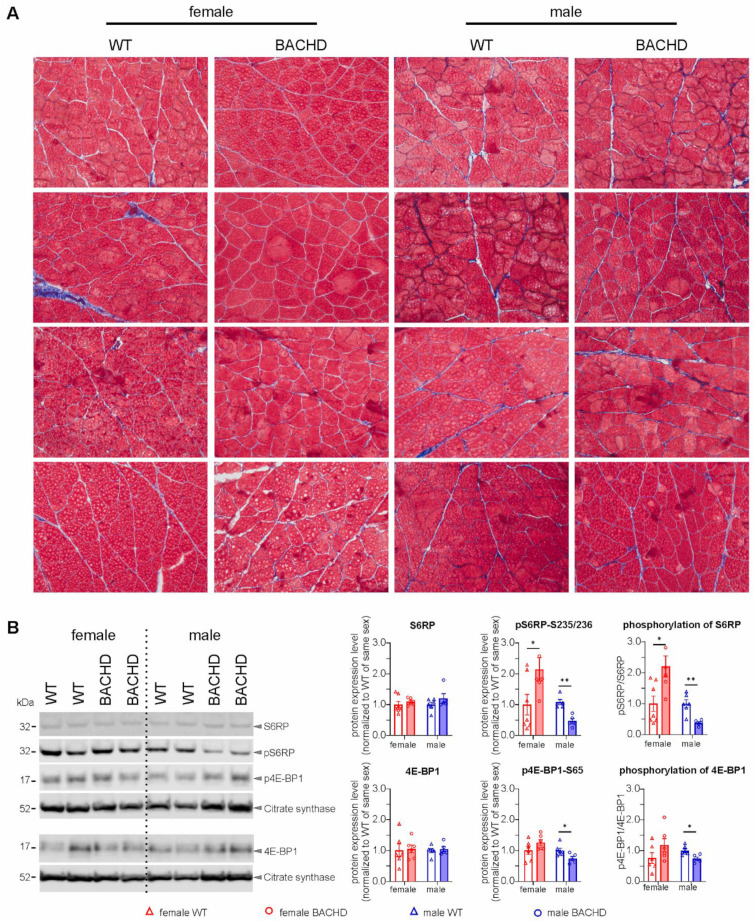
Muscle morphology and quantification of proteins regulating growth and cell proliferation. (**A**) Trichrome stain shows overall a comparable collagen tissue content (blue) and muscle fibers (red) in female or male BACHD rats compare to WT littermates of the same sex (*n* = 4/genotype/sex). (**B**) Levels of S6RP and 4E-BP1 (total protein, phosphorylated protein and phosphorylated/total protein) were compared between genotypes within each sex (female: *n* = 6/genotype; male: *n* = 5–6/genotype). Data are expressed as mean values ± SEM. Statistical significance was determined by Student’s *t*-test. * *p* ≤ 0.05; ** *p* ≤ 0.01.

**Figure 3 cells-11-03779-f003:**
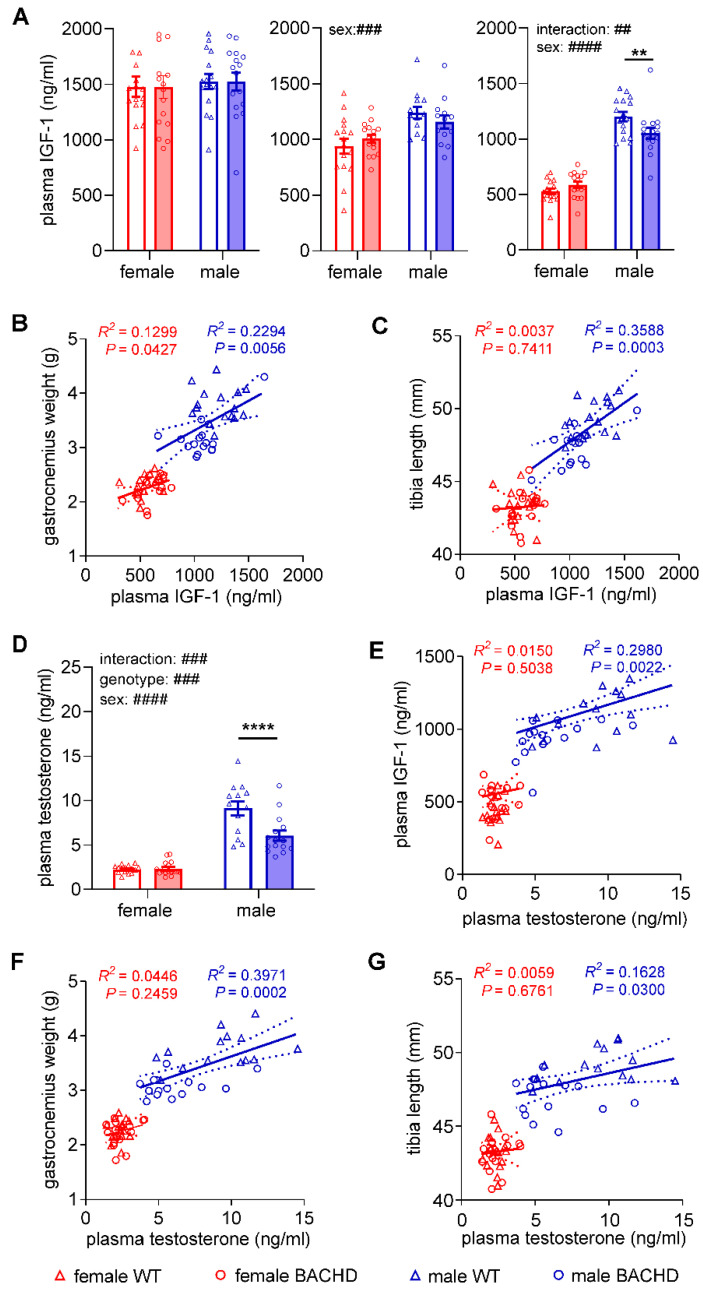
Sex-specific reductions of plasma IGF-1 and testosterone levels correlate with each other and with growth parameters in male BACHD rats at 7 months of age. (**A**) Plasma levels of IGF-1 in young (1.5 months), young adult (3 months) and middle-aged adult (7 months) groups. (**B**) Correlation between plasma levels of IGF-1 and gastrocnemius weight in female and male rats in the middle-aged group (7 months). (**C**) Correlation between plasma levels of IGF-1 and tibia length in female and male rats in the middle-aged group (7 months). Plasma levels of testosterone (**D**) in the middle-aged group (7 months). (**E**) Correlations between plasma levels of testosterone and IGF-1 (7 months). (**F**) Correlation between plasma levels of testosterone and the weight of gastrocnemius (7 months). (**G**) Correlation between plasma testosterone levels and tibia length (7 months). For all analyses *n* = 12–16/genotype/sex/age. In A and D: data are expressed as mean values ± SEM; statistical significance was determined by regular two-way ANOVA followed by Sidak’s post-hoc test. # indicates the results of two-way ANOVA analysis; * indicates the results of post-hoc test. ##/** *p* ≤ 0.01; ### *p* ≤ 0.001; ####/**** *p* ≤ 0.0001. In (**B**,**C**,**E**–**G**): data are represented as individual values (circles and triangles) with media (central continuous line) and 95% confidence interval ranges (dashed lines); statistical significance was determined by regression analysis.

**Figure 4 cells-11-03779-f004:**
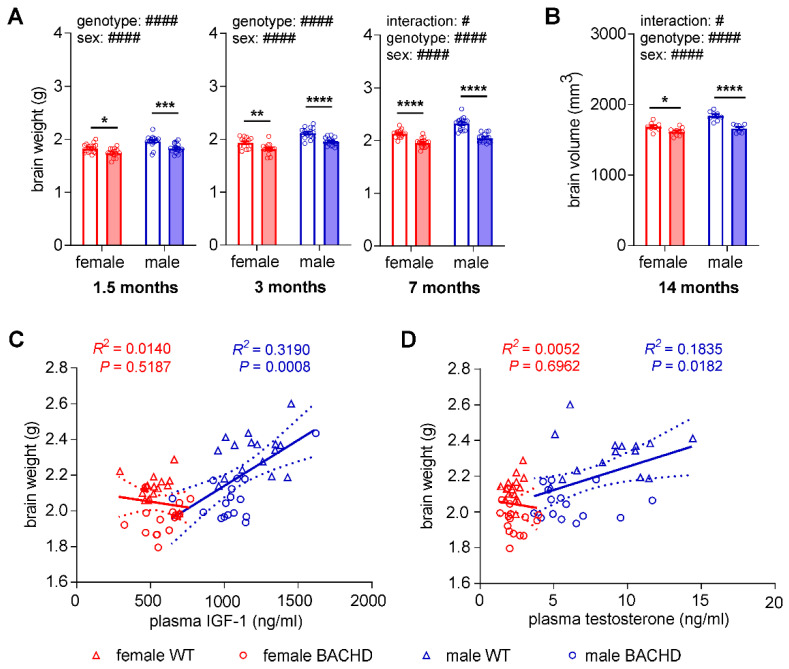
Sex-differences in brain weight and volume reductions and correlation of brain weight with plasma IGF-1 and testosterone levels in BACHD rats. (**A**) Brain weight in young (1.5 months), young adult (3 months) and middle-aged adult (7 months) (*n* = 12–16/genotype/sex/age) groups. (**B**) Whole brain volume at 14 months of age (*n* = 8 female WT, 8 male WT, 10 female BACHD, 8 male BACHD). (**C**) Correlation between plasma levels of IGF-1 and brain weight at 7 months of age in male and female rats. (**D**) Correlation between plasma levels of testosterone and brain weight at 7 months of age in male and female rats. In A and B: data are expressed as mean values ± SEM; statistical significance was determined by regular two-way-ANOVA followed by Sidak’s post-hoc test. # indicates the results of two-way-ANOVA analysis; * indicates the results of post-hoc test. #/* *p* ≤ 0.05; ** *p* ≤ 0.01; *** *p* ≤ 0.001; ####/**** *p* ≤ 0.0001. In C and D: data are expressed as individual values (circles and triangles) with media (central continuous line) and 95% confidence interval ranges (dashed lines); statistical significance was determined by regression analysis.

**Figure 5 cells-11-03779-f005:**
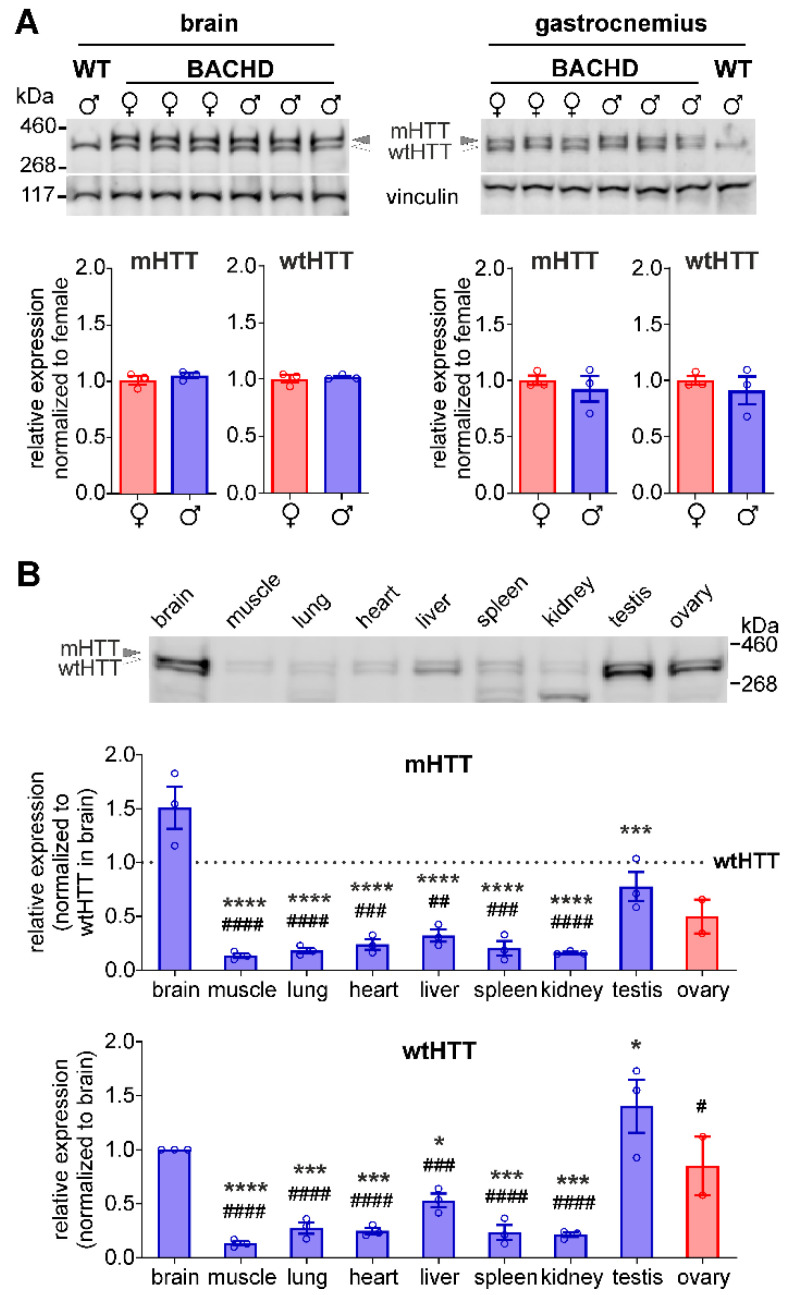
Comparisons of mHTT and wtHTT expression in females vs. males and in brain vs. other organs in the middle-aged group of BACHD rats (7 months). (**A**) Representative western blot and quantification of mHTT and wtHTT expression in the brain (left) and in the gastrocnemius muscle (right) normalized to vinculin (*n* = 3/genotype, Student’s *t*-test for each tissue). Values of mHTT and wtHTT normalized to vinculin were compared between female and male BACHD rats in brain and muscle tissue. (**B**) The expression levels of mHTT and wtHTT were compared between brain, muscle, lung, heart, liver, spleen, kidney and testis in adult male rats (*n* = 3/tissue), as well as between testis and ovary (*n* = 2 females and 3 males). 30 µg protein samples from each tissue type were loaded into the gel. Membranes were incubated with SYPRO Ruby Protein Blot Stain to detect total protein in each lane, which served as loading control for normalization. Data are expressed as mean values ± SEM. Statistical significance was determined by one-way-ANOVA followed by Dunnett’s multiple comparison test in males (brain vs. each of the peripheral organs including testis, and testis vs. each of the other peripheral organs), and *t*-test between testis and ovary. * indicates a significant difference in protein expression level between the brain and each peripheral organ; # indicates a significant difference between the testis and each of the other peripheral organs. */# *p* ≤ 0.05; ## *p* ≤ 0.01; ###/*** *p* ≤ 0.001, ####/**** *p* ≤ 0.0001.

**Figure 6 cells-11-03779-f006:**
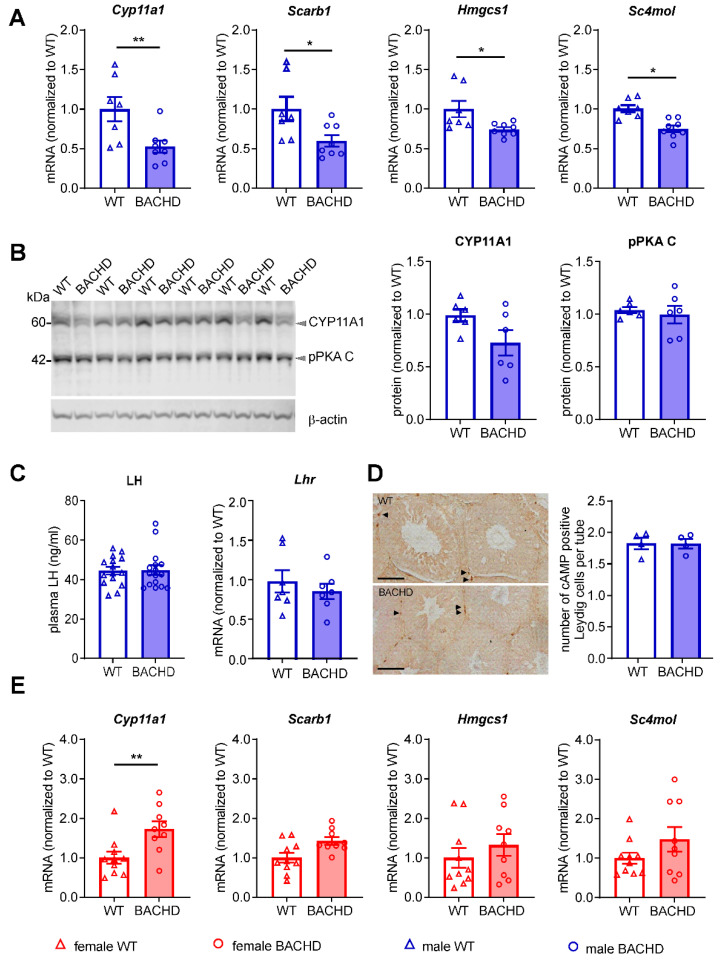
Downregulated expression of steroid biosynthesis-related genes in the testes but not ovaries of BACHD rats at 7 months of age and unchanged luteinizing hormone signaling. (**A**) mRNA expression levels of genes *Cyp11a1*, *Scarb1*, *Hmgcs1* and *Sc4mol* in the testes (*n* = 6–7 WT, *n* = 7–8 BACHD). (**B**) Representative western blot image, and quantification of protein expression levels of CYP11A1 and cAMP-activated catalytic subunit C of protein kinase A (pPKA C) in the testes (*n* = 6/genotype). (**C**) Plasma luteinizing hormone (LH) levels (*n* = 16/genotype) and mRNA expression levels of LH receptor (*Lhr*) (*n* = 7/genotype). (**D**) Number of cAMP-positive Leydig cells in the testis of WT and BACHD rats (*n* = 4/genotype) and representative images of anti-cAMP immunohistochemical staining (arrows in the images) in the testis of WT and BACHD rats (*n* = 4/genotype). (**E**) mRNA expression levels of *Cyp11a1*, *Scarb1*, *Hmgcs1* and *Sc4mol* in the ovaries (*n* = 10 WT, *n* = 9 BACHD). Data are expressed as individual values and mean ± SEM; statistical significance was determined by Student’s *t*-test for all analyses. Data are expressed as mean values ± SEM. * *p* ≤ 0.05; ** *p* ≤ 0.01.

**Figure 7 cells-11-03779-f007:**
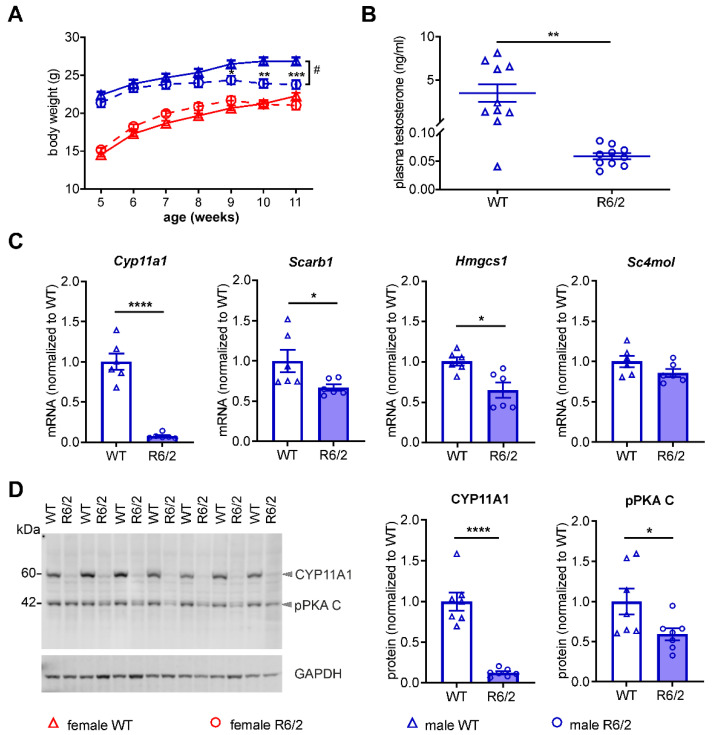
Altered body weight trajectory, plasma testosterone levels, testicular steroid biosynthesis-related gene expression and PKA activation in male R6/2 mice. (**A**) Longitudinal analysis of body weight in female and male WT and R6/2 mice from 5 to 11 weeks of age (females, *n* = 16/genotype; males, *n* = 11/genotype). (**B**) Plasma testosterone levels (*n* = 11/genotype) in 11 weeks old male WT and R6/2 mice. (**C**) mRNA expression levels of steroid biosynthesis-related genes in the testes of male R6/2 mice and WT littermates at 11 weeks of age (*n* = 6/genotype). (**D**) Western blot analysis of CYP11A1 and cAMP-activated catalytic subunit C of protein kinase A (pPKA C) in the testes of male R6/2 mice and WT littermates at 11 weeks of age (*n* = 7/genotype). Statistical significance was determined by two-way repeated measures ANOVA followed by Turkey post-hoc test in A and by Student’s *t*-test in B, C, and D. Data are expressed as individual values and mean ± SEM. * *p* ≤ 0.05; ** *p* ≤ 0.01; **** *p* ≤ 0.0001.

**Table 1 cells-11-03779-t001:** Hormone level reductions and their correlations to growth parameters in male (m) and female (f) BACHD rats.

	Reduction in BACHD Rats [*p* Value]	Correlation with Growth Parameters [*p* Value]		
		Gastrocnemius	Tibia	IGF-1
**Testosterone**	m: < 0.0001 f: > 0.5	m: = 0.0002 f: = 0.2459	m: = 0.0300 f: = 0.6761	m: = 0.0022 f: = 0.5038
**17β-estradiol**	m: > 0.5 f: > 0.5	m: = 0.0132 f: = 0.2524	m: = 0.0160 f: = 0.8190	m: = 0.9145 f: = 0.0834
**GH**	m: > 0.5 f: > 0.5	m: = 0.6540 f: = 0.3301	m: = 0.8022 f: = 0.0390	m: = 0.7858 f: = 0.3693

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
