# Peer review of "Evidences for Mutant Huntingtin Inducing Musculoskeletal and Brain Growth Impairments via Disturbing Testosterone Biosynthesis in Male Huntington Disease Animals"

_cells, 2022, doi:10.3390/cells11233779_

Round 1
Reviewer 1 Report
Yu-Taeger et al presented an interesting and important study showing sex-dependent differences in growth in two HD mouse models: BACHD rats and R6/2 mice, which represent full-length and fragment mutant huntingtin (mHTT) animal models of HD, respectively. They found that, like HD patients, these HD mice also display muscle atrophy. Moreover, the study demonstrated that this phenotype is sex-dependent and that decreased circulating levels of testosterone hormones occurs in male HD mice, similar to the decreased testosterone levels shown in male HD patients. Further, the authors provided evidence that mHTT induces growth impairment specifically in male HD animals by disturbing the transcription of testosterone biosynthesis-related genes.
The significance of the study is high. It clarified some unclear issues of why full-length HTT animal model display an over-weight phenotype whereas HD patients often show body weight loss. In fact, HD rats (at least BACHD rats) have the same muscle atrophy as HD patients, though the increased body fat can result in over-weight. Thus, mutant HTT can alter metabolic function in rodents and causes the same phenotype of musculoskeletal mass reduction as seen in HD patients. Identification of decreased circulating IGF-1 levels not only explains some HD phenotypes but also offers a potential biomarker to evaluate disease progression. Further, transcriptional dysregulation of testosterone biosynthesis-related genes is consistent with the high level of mutant HTT expression in the testis and provides new pathogenic insight. The manuscript was well written, and the data are clearly presented. The findings should be highly interesting to the field.
I only have some minor issues for the authors to address.
1. Fig. 4B, it would be important to spell out how HTT was quantified in figure legend. This is because it is important to note that both the brain and testis express the higher level of mutant HTT than other tissues do.
2. Fig. 6D, the levels of cAMP-activated subunit C of PKA in testis were lower in R6/2 mice, which is different from BACHD rats. The authors need to discuss this difference. Would it be due to different forms of mutant HTT in the testis or due to species related difference between mouse and rat?
Author Response
Dear reviewer,
We thank the you for your helpful comments. We addressed all your points and believe that these further strengthened our manuscript. We sincerely hope that the revised manuscript meets your high standards.
Please find our response to the comments point by point below. All changes to the manuscript are highlighted in the new version.
Yours sincerely,
Prof. Dr. Med. Huu Phuc Nguyen
- Fig. 4B, it would be important to spell out how HTT was quantified in figure legend. This is because it is important to note that both the brain and testis express the higher level of mutant HTT than other tissues do.
We thank the reviewer for this comment. We have added the requested information explaining how HTT was quantified in all tissues. The changes can be found in the legend of Fig. 5B (previous Fig. 4B).
- Fig. 6D, the levels of cAMP-activated subunit C of PKA in testis were lower in R6/2 mice, which is different from BACHD rats. The authors need to discuss this difference. Would it be due to different forms of mutant HTT in the testis or due to species related difference between mouse and rat?
The reviewer is right. We have now discussed the differences in change of cAMP-activated subunit C of PKA between BACHD rats and R6/2mice. This information has been reported in the discussion of the manuscript.
Reviewer 2 Report
The authors mainly studied the tissue/body weights of male and female BACHD rats versus the controls, and claimed that “Mutant Huntingtin Impairs Musculoskeletal and Brain Growth via Disturbing Testosterone Biosynthesis in Male Huntington Disease Animals”. This conclusion per se is significant and novel, but it is not demonstrated by the authors’ experimental results. There are a number severe logic flaws that need to be addressed.
Major concerns:
1. The measurement of musculoskeletal and brain growth are not convincing. The tissue weights were used as the major readout, without any cellular or molecular marker for cell proliferation and differentiation. The basic physiological/pathophysiological characterization of the brain and muscle tissues are missing. There were even no neuron or muscle fiber morphology detection. One can never draw conclusions on impairment of growth or atrophy just based the tissue weight measurements. On top of this, there was not blinding or randomization reported, and thus even the tissue dissection and weight measurements could be easily biased. Thus, the conclusion on the impairment of “Musculoskeletal and Brain Growth” is not sufficiently demonstrated.
2. The potential contribution of atrophy was also not tested, but simply excluded based on appearance of tissue weight difference at the age of 7 months or earlier. This is clearly flawed, because atrophy may occur before 7 months as well.
3. The authors claimed that mutant HTT causes disturbed testosterone biosynthesis, which then causes musculoskeletal and brain growth impairment, but the authors only provided correlative data. In fact, the muscle/brain tissue growth may easily be directly affected by mHTT expressed in these tissues per se, which may also be influenced by sex-specific factors that are different in males and females. In addition, the metabolic differences in male and females such as food/energy intake and exercise may also influence the changes of tissue weights in HD animals. All these influencing factors need to be tested. In addition, to establish the causal relationship, the authors need to modulate mutant HTT levels in the testis and/or testosterone levels to test their impacts.
4. The models used in the study were also problematic. This study was inspired by the weight loss in HD patients in both males and females, but the BACHD rats only exhibit very mild weight loss in the males but not females. In addition, the BACHD mice actually exhibit larger weights compared to control mice (PMID: 18550760). This questions whether the BACHD transgene is the fundamental cause of body and tissue weight changes.
Minor concerns:
1. The introduction lacked relevant papers studying the HD muscle phenotypes (for example, the ones from Gill Bates’ group) and the HD developmental deficits (for example, the ones from Sandrine Humbert’s group).
2. The statistical analyses were unclear. For example, F values of ANOVA tests were not shown, and the type of post-hoc tests were not given. The power analyses to justify the sample size were also missing.
Author Response
Dear reviewer,
We thank you for your helpful comments and discussion. We addressed all your points and believe that these further strengthened our manuscript. We sincerely hope that the revised manuscript meets your high standards.
Please find our response to the comments point by point below. All changes to the manuscript are highlighted in the new version.
Yours sincerely,
Prof. Dr. Med. Huu Phuc Nguyen
Comments and Suggestions for Authors
Major concerns:
- The measurement of musculoskeletal and brain growth are not convincing. The tissue weights were used as the major readout, without any cellular or molecular marker for cell proliferation and differentiation. The basic physiological/pathophysiological characterization of the brain and muscle tissues are missing. There were even no neuron or muscle fiber morphology detection. One can never draw conclusions on impairment of growth or atrophy just based the tissue weight measurements. On top of this, there was not blinding or randomization reported, and thus even the tissue dissection and weight measurements could be easily biased. Thus, the conclusion on the impairment of “Musculoskeletal and Brain Growth” is not sufficiently demonstrated.
We thank the reviewer for these comments. We agree with the reviewer’s point and have now performed analyses of both morphology and apoptosis markers in the brain and gastrocnemius muscle of BACHD rats and WT littermates. Brain analyses were performed in 9 months old rats, and muscle analyses were carried out in 7 months old (middle-aged) rats that already present brain and muscle weight changes. The results of our analysis suggest no ongoing atrophy in the BACHD rat brain or muscle. Furthermore, we quantified the levels of phosphorylated S6RP and 4E-BP1, the two most established proteins involved in protein synthesis and cell proliferation and therefore in the control of the growth process. In line with the results of growth parameters, S6RP and 4E-BP1 show significant phosphorylation changes in BACHD rats. S6RP phosphorylation was increased and 4E-BP1 phosphorylation was unchanged in females, while phosphorylation of both proteins was decreased in males. These findings support the conclusion that the reduction in musculoskeletal and brain mass in BACHD rats is dependent on growth impairment rather than organ atrophy. The new results can be found in the manuscript results section 3.2 (analyses in gastrocnemius) and 3.4 (analyses of neurodegeneration and apoptosis events in brain).
We apologize for not having clarified that all experiments in our manuscript were performed by an experimenter blind to the experimental groups, and that during each measurement, animals of different groups were randomized according to age, sex and genotype. We have now added this statement in the materials and methods section.
- The potential contribution of atrophy was also not tested, but simply excluded based on appearance of tissue weight difference at the age of 7 months or earlier. This is clearly flawed, because atrophy may occur before 7 months as well.
We have addressed this point in the reply to the reviewer major concern 1. Please see the reply to major concern 1.
- The authors claimed that mutant HTT causes disturbed testosterone biosynthesis, which then causes musculoskeletal and brain growth impairment, but the authors only provided correlative data. In fact, the muscle/brain tissue growth may easily be directly affected by mHTT expressed in these tissues per se, which may also be influenced by sex-specific factors that are different in males and females. In addition, the metabolic differences in male and females such as food/energy intake and exercise may also influence the changes of tissue weights in HD animals. All these influencing factors need to be tested. In addition, to establish the causal relationship, the authors need to modulate mutant HTT levels in the testis and/or testosterone levels to test their impacts.
We agree that it would be important to test the influencing factors mentioned by the reviewer and that modulating mutant HTT levels / testosterone levels in the testis would be determinant to finally establish a causal relationship between mutant HTT and the observed brain and muscle changes in the BACHD rat model. While these analyses will be performed in future studies, we have mentioned these points in the discussion of our revised manuscript. Furthermore, we have changed the title of our manuscript in order to better reflect our results. The new title reads: “Evidences for mutant huntingtin inducing musculoskeletal and brain growth impairments via disturbing testosterone biosynthesis in male Huntington disease animals”.
- The models used in the study were also problematic. This study was inspired by the weight loss in HD patients in both males and females, but the BACHD rats only exhibit very mild weight loss in the males but not females. In addition, the BACHD mice actually exhibit larger weights compared to control mice (PMID: 18550760). This questions whether the BACHD transgene is the fundamental cause of body and tissue weight changes.
We thank the reviewer for rising this important point. Two main factors contributing to body weight are fat mass and fat free mass (lean mass). In male BACHD rats, we observe a decreased muscle weight and bone length that are representative of a decreased lean mass which is also observed in patients. Both female and male HD rats show an increased fat mass for which the relation to the human HD phenotype is questionable. Such increased fat mass may compensate for the reduced musculoskeletal weight in males leading to a normal or slightly altered body weight, and may result in an even increased body weight in females where the musculoskeletal weight is unchanged. Nevertheless, body weight is a complex readout and two main factors determining it are lean mass and fat tissue. In the present study we focused our analyses on muscle weight and bone length which are related to lean mass and more relevant to the disease in patients. As the changes observed in the BACHD rats in this study may have been specific for this model / transgene construct, we confirmed the findings also in R6/2 mice which differently than BACHD rats (i) represent a model carrying a fragment mHTT and (ii) is generated in a different animal species.
Minor concerns:
- The introduction lacked relevant papers studying the HD muscle phenotypes (for example, the ones from Gill Bates’ group) and the HD developmental deficits (for example, the ones from Sandrine Humbert’s group).
We thank the reviewer for pointing this out. We have added the relevant literature on HD muscle phenotypes and on HD developmental deficits in the introduction session according to the reviewer’s suggestions.
- The statistical analyses were unclear. For example, F values of ANOVA tests were not shown, and the type of post-hoc tests were not given. The power analyses to justify the sample size were also missing.
We have added the F values of the ANOVA tests to the result section of the manuscript. We have also specified the types of post-hoc test in material and methods and figure legends.
Reviewer 3 Report
I suggest you summarize what is known in the BAC HD mouse model - in the context of male vs female body weight, adipose and muscle weight. There appear to be differences between the mouse and rat BAC HD models in these measures. Has BAC HD mouse been extensively characterized for plasma IGF1, and testosterone as well as WAT and muscle weight? Does species (mouse vs rat) give disparate results?
In the R6/2 data - no differences in BW for female transgenics was seen. This contrasts with a lot of published data. Please address. In this study, was plasma IGF1 measured?
Please also comment what is known or not known about WAT, muscle weight and BW effects in other HD mouse models (i.e. YAC128, Q175, Q111). Are your results generally translatable across only some or all of the models?
Author Response
Dear reviewer,
We thank you for your helpful comments. We addressed all your points and believe that these further strengthened our manuscript. We sincerely hope that the revised manuscript meets your high standards.
Please find our response to the comments point by point below. All changes to the manuscript are highlighted in the new version.
Yours sincerely,
Prof. Dr. Med. Huu Phuc Nguyen
Comments and Suggestions for Authors
I suggest you summarize what is known in the BAC HD mouse model - in the context of male vs female body weight, adipose and muscle weight. There appear to be differences between the mouse and rat BAC HD models in these measures. Has BAC HD mouse been extensively characterized for plasma IGF1, and testosterone as well as WAT and muscle weight? Does species (mouse vs rat) give disparate results?
We thank the reviewer for this comment. Body weight has been analyzed separately in females and males in many HD rodent models. Previous studies suggested that body weight in HD rodents correlates with the expression levels of full-length HTT. In particular, HD rodent models carrying more copies of full-length HTT such as BACHD mice and YAC mice, display increased body weight [1-4], whereas models expressing a fragment of mHTT or HD knock-in models show decreased body weight due to the depletion of full-length wild type HTT [5-9]. Regardless of increased or decreased body weight, many HD rodent models show sex differences in the body weight phenotype, and this has been observed across rat and mouse models carrying different transgenic constructs. BACHD females show more prominent body weight gain compared to BACHD males, while a more pronounced body weight loss has been observed in HD males compared to females in tg51 rats and R6/2, HdHQ111, zQ175 and CAGKI mice [7, 8, 10-14], as well as in a humanized mouse model of HD [9]. We have added all this information on body weight in the introduction of the manuscript in order to make our study background clearer.
There is very limited data on muscle and WAT weight in HD rodent models, and data comparing these parameters between sexes is especially scarce. One study analyzed IGF-1 in BACHD and YAC128 mice, and showed increased IGF-1 in HD mice compared with WT controls in both models. The study, though, used mixed sex experimental groups.
In the R6/2 data - no differences in BW for female transgenics was seen. This contrasts with a lot of published data. Please address. In this study, was plasma IGF1 measured?
In our study, body weight in R6/2 mice (carrying 140 CAG repeats) was analyzed in the presymptomatic/ early symptomatic phase. Our results on body weight in 5 -11 weeks old mice are in line with several published studies showing comparable weight gain in R6/2 mice and WT littermates from 5 to 8 weeks and a trend towards reduced body weight in R6/2 mice at 11 weeks. Two-way ANOVA analyses didn’t reveal a genotype effect in females. For example, one study reported that female R6/2 mice start to lose body weight at 11 weeks [8], and another study in female mice, reported that the body weight of R6/2 increased at a similar rate as WT until 10 weeks of age, and then gradually declined from 12 weeks [15].
We agree that measuring IGF-1levels besides testosterone in R6/2 mice would be very informative in this study. Unfortunately, we have not been able to measure IGF-1levels in R6/2 mice due to the limited volume of our blood samples, but these analyses will be performed in a follow up study.
Please also comment what is known or not known about WAT, muscle weight and BW effects in other HD mouse models (i.e. YAC128, Q175, Q111). Are your results generally translatable across only some or all of the models?
We thank the reviewer for this point. We have discussed the requested parameters in our reply to the first comment of the reviewer.
References
- Gray, M., et al., Full-length human mutant huntingtin with a stable polyglutamine repeat can elicit progressive and selective neuropathogenesis in BACHD mice. J Neurosci, 2008. 28(24): p. 6182-95.
- Hult, S., et al., Mutant huntingtin causes metabolic imbalance by disruption of hypothalamic neurocircuits. Cell Metab, 2011. 13(4): p. 428-439.
- Pouladi, M.A., et al., Full-length huntingtin levels modulate body weight by influencing insulin-like growth factor 1 expression. Hum Mol Genet, 2010. 19(8): p. 1528-38.
- Van Raamsdonk, J.M., et al., Body weight is modulated by levels of full-length huntingtin. Hum Mol Genet, 2006. 15(9): p. 1513-23.
- Brooks, S., et al., Longitudinal analysis of the behavioural phenotype in YAC128 (C57BL/6J) Huntington's disease transgenic mice. Brain Res Bull, 2012. 88(2-3): p. 113-20.
- Dickson, E., et al., Hypothalamic expression of huntingtin causes distinct metabolic changes in Huntington's disease mice. Mol Metab, 2022. 57: p. 101439.
- Menalled, L., et al., Systematic behavioral evaluation of Huntington's disease transgenic and knock-in mouse models. Neurobiol Dis, 2009. 35(3): p. 319-36.
- Phan, J., et al., Adipose tissue dysfunction tracks disease progression in two Huntington's disease mouse models. Hum Mol Genet, 2009. 18(6): p. 1006-16.
- Southwell, A.L., et al., A fully humanized transgenic mouse model of Huntington disease. Hum Mol Genet, 2013. 22(1): p. 18-34.
- Bode, F.J., et al., Sex differences in a transgenic rat model of Huntington's disease: decreased 17beta-estradiol levels correlate with reduced numbers of DARPP32+ neurons in males. Hum Mol Genet, 2008. 17(17): p. 2595-609.
- Chen, J.Y., et al., Effects of the Pimelic Diphenylamide Histone Deacetylase Inhibitor HDACi 4b on the R6/2 and N171-82Q Mouse Models of Huntington's Disease. PLoS Curr, 2013. 5.
- Dorner, J.L., et al., Sex differences in behavior and striatal ascorbate release in the 140 CAG knock-in mouse model of Huntington's disease. Behav Brain Res, 2007. 178(1): p. 90-7.
- Menalled, L.B., et al., Genetic deletion of transglutaminase 2 does not rescue the phenotypic deficits observed in R6/2 and zQ175 mouse models of Huntington's disease. PLoS One, 2014. 9(6): p. e99520.
- Peng, Q., et al., Characterization of Behavioral, Neuropathological, Brain Metabolic and Key Molecular Changes in zQ175 Knock-In Mouse Model of Huntington's Disease. PLoS One, 2016. 11(2): p. e0148839.
- Carter, R.J., et al., Characterization of progressive motor deficits in mice transgenic for the human Huntington's disease mutation. J Neurosci, 1999. 19(8): p. 3248-57.
Round 2
Reviewer 2 Report
The major claim is that the muscle/brain defects in HD are likely due to abnormal tissue growth rather than tissue atrophy. This claim is actually contradictory to the prevailing view in the HD field and needs to be made carefully. For the muscle, Bate's group alone has published multiple papers demonstrating atrophgy (PMID 32306066, 29079832, 25748626, etc.). For the brain, there are hundreds of paper demonstrating atrophy in animal models and human patients. While the authors did present some additional phenotypic data, a robust demonstration of growth rather than atrophy is still lacking. Muscle stem cell number and differentiation were not tested. Cell proliferation ability and cell cycle markers were not tested. No staining for proliferation cells using BrdU, etc. The major molecular demonstration was based on S6RP and 4E-BP1, which are actually mRNA translation regulators that could be influenced by many factors especially the AKT pathway that is also involved in muscle atrophy (PMID 32858949). Indeed, their changes in the females were not consistent with the prediction direction and the reasons were unclear.